# Chemical Fingerprint Analysis and Quantitative Analysis of Saccharides in Morindae Officinalis Radix by HPLC-ELSD

**DOI:** 10.3390/molecules26237242

**Published:** 2021-11-29

**Authors:** Hongmei Sun, Yini Cai, Jie Shen, Enyao Ma, Zhimin Zhao, Depo Yang, Xiuwei Yang, Xinjun Xu

**Affiliations:** 1Academy for Advanced Interdisciplinary Studies, Peking University, Beijing 100871, China; sunhongmei2010@126.com; 2Infinitus (China) Co., Ltd., Jiangmen 529100, China; 3School of Pharmaceutical Sciences, Sun Yat-sen University, Guangzhou 510006, China; caiyn5@mail2.sysu.edu.cn (Y.C.); shenjie2901@sina.com (J.S.); zhaozhm2@mail.sysu.edu.cn (Z.Z.); lssydp@mail.sysu.edu.cn (D.Y.); 4Guangzhou Caizhilin Pharmaceutical Co., Ltd., Guangzhou 510360, China; maenyao@163.com

**Keywords:** Morindae officinalis radix, HPLC-ELSD, saccharides, TIM-TOF-MS, fingerprint

## Abstract

A method based on high performance liquid chromatography with evaporative light scattering detection (HPLC-ELSD) was developed for the quantitative analysis of three active compounds and chemical fingerprint analyses of saccharides in Morindae officinalis radix. Ten batches of Morindae officinalis radix were collected from different plantations in the Guangdong region of China and used to establish the fingerprint. The samples were separated with a COSMOIL Sugar-D column (4.6 mm × 250 mm, 5 μm) by using gradient elution with water (A) and acetonitrile (B). In addition, Trapped-Ion-Mobility (tims) Time-Of-Flight (tims TOF) was used to identify saccharides of Morindae officinalis radix. Fingerprint chromatogram presented 26 common characteristic peaks in the roots of *Morinda officinalis* How, and the similarities were more than 0.926. In quantitative analysis, the three compounds showed good regression (r = 0.9995–0.9998) within the test ranges, and the recoveries of the method were in the range of 96.7–101.7%. The contents of sucrose, kestose and nystose in all samples were determined as 1.21–7.92%, 1.02–3.37%, and 2.38–6.55%, respectively. The developed HPLC fingerprint method is reliable and was validated for the quality control and identification of Morindae officinalis radix and can be successfully used to assess the quality of Morindae officinalis radix.

## 1. Introduction

*Morinda officinalis* How (Rubiaceae) is one of the most popular medicinal plants in southern regions in China, such as Guangdong, Fujian and Guangxi provinces [1]. In particular, Guangdong is the main producing area of the genuine medicinal materials of Morindae officinalis radix. The produce from geo-authentic producing areas has higher quality than others; therefore, it is widely cultivated in Guangdong Province to meet the high demand [2]. Morindae officinalis radix is mainly used for the treatment of impotence, osteoporosis, depression, and inflammation [2,3,4]. Oligosaccharides, polysaccharides, iridoids, and anthraquinones are the main active components of Morindae officinalis radix [5]. Among them, oligosaccharides and polysaccharides exhibit antioxidant [6], anti-post-traumatic stress disorder [7], antidepressant [8], anti-Alzheimer [9], and antiosteoporosis activities [10]. In addition, oligosaccharides including bajijiasu, sucrose, nystose, fructofuranosylnystose, inulintype hexasaccharide, inulintype heptasaccharide, inulintype octasaccharide, and inulintype enneasaccharide have been isolated from Morindae officinalis radix by size-exclusion chromatography [11,12]. The characterization of inulin-type fructooligosaccharides with different degrees of polymerization (DP) were studied by nuclear magnetic resonance and mass spectra; the structures were identified as fructose unit chains linked by β-(2→1)-D-fructosyl-fructose bonds and a glucose terminus linked through an α-(1→2)-D-glucopyranosyl bond [12,13]. In 2012, fructooligosaccharides (DP3-DP9) of Morindae officinalis radix received permission to be prescribed for the treatment of depression from the Chinese Food and Drug Administration [14].

It is important to determine saccharides for Morindae officinalis radix to control their quality. Owing to the lack of chromophores, saccharides cannot be detected by ultraviolet detection [14]. Recently, high performance liquid chromatography (HPLC) with refractive index detection (RID) and evaporative light scattering detection (ELSD) and high-performance anion exchange chromatography (HPAEC) have been used to determine fructooligosaccharides [13,15,16]. HPAEC was conducted by specific instruments; HPLC-RID was inapplicable for gradient elution and was affected by ambient temperature [17]. ELSD has been widely used to determine compounds without chromophores. In Chinese Pharmacopoeia, nystose was used to evaluate the quality of Morindae officinalis radix by HPLC-ELSD [18]. However, an index component was insufficient for the quality assessment. Therefore, it is necessary to establish a comprehensive and systematic standard to evaluate the quality of Morindae officinalis radix. HPLC fingerprint was an effective method and had been used for the quality assessment of Kaempferia galangal [19], Centella asiatica [20], and Gentiana crassicaulis [21], etc. 

Ma and his partners established the HPLC-ELSD fingerprint analysis method to study carbohydrate constituents in Morindae Officinalis Radix and its salt-processed products. HPLC analysis was performed on an Agilent ZORBAX carbohydrate analysis column (4.6 mm × 250 mm, 5 μm), which is a traditional aminopropyl column commonly used for the separation of monosaccharides and disaccharides. As a matter of fact, some sugars are adsorbed on the stationary phase, causing tailing or even no elution when using this column to separate some certain sugars (arabinose or galactose), which is confirmed in the results of this paper. Only sixteen co-possessing peaks with the tail peak problem were selected as the fingerprint peaks of Morindae Officinalis Radix by taking the nystose peak as the referential peak. In addition, other more prominent peaks in the fingerprint have not yet been identified [22]. A COSMOIL Sugar-D column (4.6 mm × 250 mm, 5 μm) filled with a multipoint bonded aminopropyl filler was used in this study due to its superior performance in the separation of saccharides. To our knowledge, the chemical fingerprint analysis of saccharides in *Morinda officinalis* How roots by using this new type of sugar-based column has not yet been reported. Hence, the scope of the work included the development and validation of the HPLC-ELSD method for determining sucrose (C_12_H_22_O_11_), kestose (C_18_H_32_O_16_), and nystose (C_24_H_42_O_21_) in plant roots. The extraction procedure, supported by an orthogonal array design, was also optimized by investigating several experimental parameters that influenced the extraction yields. Ten batches of Morindae officinalis radix were collected from Guangdong Province. Saccharides from Morindae officinalis radix were identified through tims Time-Of-Flight (tims TOF) and analyzed with the developed method.

## 2. Results

### 2.1. Optimization of Extraction Conditions

In order to obtain optimum extraction efficiency, variables involved in the whole extraction procedure, including extraction time, extraction temperature, solvent-solid ratio, and ethanol concentration, were investigated. These variables were evaluated by single-factor experiments for the yields of sucrose, kestose, and nystose. Extraction time (1, 2, 3, 4, and 5 h) and extraction temperature (60, 70, 80, 90, and 100 °C) together with the solvent–solid ratio (10, 15, 20, 25, and 30 mL/g) were optimized for the sake of environmentally friendly operation and to save resources (Figure 1A–C). We can see the yields of three components increased until the maximum yields were reached at the time of 3 h, temperature of 90 °C, and solvent–solid ratio of 25 mL/g. In addition, a series of aqueous ethanol solutions at different concentrations (30, 40, 50, 60, 70, 80, and 90%) were compared. We can see that the 50% aqueous solution of ethanol obtained the highest yield (Figure 1D). The results showed that extraction time had a modest effect; therefore, other conditions were further optimized by orthogonal array design with the extraction time fixed as 3 h.

The orthogonal array design was conducted with three factors and three levels (Table 1); nine sets of experiments were conducted in a random design (Table 2). The yields of sucrose, kestose, and nystose were transformed into the overall desirability. The effects of extraction conditions on the contents were as follows: temperature (A) > ethanol concentration (C) > solvent–solid ratio (B) (R_A_ > R_C_ > R_B_). The optimal combination was A_2_B_3_C_3_: a temperature of 90 °C, solvent–solid ratio of 30 mL/g, and ethanol concentration of 60%. Under the optimal condition, the contents of sucrose, kestose, and nystose were 1.67 ± 0.01%, 0.79 ± 0.02%, and 2.13 ± 0.01%, respectively.

The commonly used extraction methods of saccharides in Morindae Officinalis Radix include water extraction and ethanol aqueous solution reflux extraction. It was confirmed that oligose was almost completely hydrolyzed at pH 2–3 but the hydrolyzation was alleviated at pH 3–4, with many kinds of new unknown compositions being generated. The oligosaccharides became stable when the pH was between 6 and 10. We easily found that Morinda officinalis oligosaccharides is unstable in water. Additionally, beyond that, some polysaccharides can also be extracted in water, so it is necessary to add ethanol to precipitate polysaccharides after using a water extraction method, while this step can be omitted when using the ethanol aqueous reflux extraction method. Thus, ethanol aqueous solution reflux extraction was chosen to be optimized in this study. When Morinda officinalis oligosaccharides samples were extracted with different percentages of ethanol, they only hydrolyzed when ethanol was in the range of 10–30%, with a small unknown composition. With that in mind, a series of aqueous ethanol solutions at different concentrations (30, 40, 50, 60, 70, 80, and 90%) were compared to optimize the extraction conditions. We found that the saccharides are stable in ethanol solutions when the concentration is over 30% [23].

The orthogonal array design is an effective and efficient method to solve multi-factor and multi-level problems. It determines the experimental design through a set of orthogonal tables, which are characterized by regularity and comparability. Each factor level can be evenly dispersed and can better eliminate the impact of different units and dimensions on the results with high accuracy and reliability. The orthogonal array design can greatly reduce the number of tests and is often used to investigate the main effect of a series of single factors, which means it is suitable for optimization with few factor levels and unclear interaction [24]. In the manuscript, we identified three levels of factors to examine through single-factor tests. On this basis, the optimal extraction conditions were obtained by the orthogonal array design faster and with more practical results.

The total oligosaccharide content obtained by ultraviolet spectrophotometry was used as the evaluation index in the current study regarding the extraction of saccharides in Morindae Officinalis Radix [25,26]. The specificity of this method is poor because monosaccharides, oligosaccharides and polysaccharides all have the same derivatization reaction. The results obtained by ultraviolet spectrophotometry were greater than the actual oligosaccharide content. The technique of taking the contents of sucrose, kestose, and nystose obtained by HPLC-ELSD as evaluation indexes to optimize the extraction process of the saccharides in Morindae Officinalis Radix has not yet been reported.

### 2.2. Optimization of HPLC Condition

HPLC conditions were optimized to obtain the desired resolution. Different mobile phases, including water–methanol, water–acetonitrile, and triethylamine (0.1%, *v*/*v*)–acetonitrile, were tested using isocratic elution and gradient elution. Good resolution and symmetric peak shape were obtained when water–acetonitrile was used; the final gradient was confirmed accordingly. Triethylamine did not need to be added as a tailing agent in the mobile phase due to the fact that the column selected showed excellent performance in carbohydrate analysis with no peak tailing. The determined gradient conditions shortened the analysis time as much as possible on the basis of a good separation of each peak. The signal intensity and baseline noise of the detected components peaks at different drift tube temperatures (35, 40, 50, 60, and 70 °C) and different gains (4, 5, and 6) were compared. The drift tube temperature determined was 35 °C, and the gain determined was eventually 4. Lower drift tube temperature (105 °C in Ma’s study) can avoid the decomposition of the components, which could result in response reduction. Significantly, a mixture of methanol and acetonitrile and water was used as the mobile phase in tangled and complicated gradient modes in Ma’s study, which means specific instruments with a four-element pump were required, and poor repeatability of the experiment was almost inevitable. In order to overcome the disadvantage of long elution times in traditional aminopropyl columns, the flow rate was set at 1.2 mL/min in Ma’s study [22]. Thus, higher requirements were put forward with regard to the response speed of the detector. The COSMOIL Sugar-D column has a shorter elution time in comparison to the traditional aminopropyl column, achieving the application of commonly used elution speed (1.0 mL/min) in the analysis. The typical HPLC chromatograms profile of three standards peaks are shown in Figure 2. Sucrose, kestose, and nystose in sample solutions were eluted separately, and no interfering substances were found in the chromatograms of the blank and sample solutions.

### 2.3. Method Validation of Quantitative Analysis

The method was validated in terms of system adaptability, linearity, LOD, LOQ, repeatability, stability, and recovery test. Regression data, LODs, and LOQs for three standard substances are given in Table 3. The system adaptability was performed by six replicate determinations of a sample solution. The calibration curves were constructed by plotting the log of peak areas versus the log of concentration of each compound. The LODs and LOQs were evaluated as S/N of 3 and 10, respectively. Regression data, LODs, and LOQs for three standard substances are given in Table 2. The repeatability test was performed by six replications of a batch. Moreover, the sample solution was found to be stable within 12 h. The recovery test was determined by the standard addition method. Samples were prepared at three concentration levels in triplicate by spiking known quantities of each of the three standards (50%, 100%, and 150% of the sample content) into the sample, which contained half the normal amount of Morindae officinalis radix powdered samples in “3.4.”, and then were extracted and analyzed according to the procedures in “3.4.”. The calculation formula of recovery is as follows:(1)$$\mathrm{Recovery}\;(\%)=\frac{C-A}{B}*100\%$$

A, B, and C are the measured component content of the sample, the content of the standards added, and the measured value, respectively.

The validation data are shown in Table 4. The results of method validation indicated that the HPLC-ELSD method was accurate for use in simultaneous determinations of sucrose, kestose, and nystose.

### 2.4. Analysis of Tims TOF

The total ion chromatogram with a negative ion mode by tims TOF was more sensitive than that with a positive ion mode, which used to detect saccharides from Morindae officinalis radix, and 19 peaks were detected (Figure 3). The [M-H]^−^ were obtained at *m*/*z* 341, 503, 665, 827, 989, 1151, 1313, 1475, 1637, and 1799 (Table 5). In addition, saccharides with high molecular weights were ionized as [M-2H]^2−^ at *m*/*z* 980, 1061, 1142, 1223, 1304, 1385, and 1466. There was a 162 (C_6_H_10_O_5_, hexoses) difference in the molecular weight between two neighbor oligosaccharides. Comparing the accurate molecular weights with these of the reported oligosaccharides in this plant [12], the compounds were deduced as sucrose (peak 1), kestose (peak 2), nystose (peak 3), fructofuranosylnystose (peak 4), fructooligosaccharide DP6 (peak 6), fructooligosaccharide with DP7 (peak 8), fructooligosaccharide DP8 (peak 9), fructooligosaccharide DP9 (peak 10), fructooligosaccharide DP10 (peak 11), fructooligosaccharide DP11 (peak 12), fructooligosaccharide DP12 (peak 13), fructooligosaccharide DP13 (peak 14), fructooligosaccharide DP14 (peak 15), fructooligosaccharide DP15 (peak 16), fructooligosaccharide DP16 (peak 17), fructooligosaccharide DP17 (peak 18), and fructooligosaccharide DP18 (peak 19).

### 2.5. Method Validation of HPLC Fingerprint Analysis

The fingerprints method was validated in terms of system adaptability, repeatability, and stability, according to the Similarity Evaluation System for Chromatographic Fingerprint of Traditional Chinese Medicine. System adaptability was evaluated by injecting the sample solution six times; their similarities were more than 0.995. The repeatability tests were performed with six independently sample solutions; their similarities were more than 0.999. The stability was obtained by injecting the same sample solution stored at 0, 2, 4, 6, 8, 10, and 12 h after the preparation; their similarities were more than 0.997. These results indicated that the method can be used in fingerprint analysis.

### 2.6. HPLC Fingerprint Analysis

The HPLC fingerprints of Morindae officinalis radix were analyzed to generate the reference fingerprint (Figure 4). Twenty-six common peaks were observed in all fingerprints, which were favorably separated under the given HPLC conditions. Compared with the standard solution, three peaks were identified as sucrose (4), kestose (6), and nystose (8). The peak of nystose was selected as the reference peak. The similarity was evaluated based on the correlation coefficient of the fingerprints. The correlation coefficients of the cortexes were 0.969, 0.986, 0.985, 0.993, 0.990, 0.997, 0.991, 0.993, 0.993, and 0.998. The contents of sucrose in all samples ranged from 1.21% to 4.56%, the contents of kestose ranged from 1.02% to 2.44%, and the contents of nystose ranged from 2.38% to 4.99% (Table 6).

### 2.7. Principal Component Analysis (PCA)

Principal component analysis (PCA) is a multivariate statistical method used to examine the correlation between multiple variables. It is used to study how to interpret the internal structure between multiple variables by using a few principal components, which are derived from the original variables, while preserving information about the original variables as much as possible. It was used to analyze the HPLC chromatographic fingerprints of 10 Morindae officinalis radix samples. The relative peak areas (RPAs) of the 26 characteristic common peaks were set as variables, while 10 batches of samples were set as observations. 

Taking the eigenvalue >1 as the standard, the eigenvalue of the first three principal component factors is 16.429, 5.215, and 1.813, respectively. PC1 explained 65.7% of the total variance in the data set, while PC2 explained 20.8%, PC3 explained 7.3%, and the contributing rate of cumulative sums of squares was 93.824% > 85% (Table 6). According to the component diagram, PC1 shows a positive correlation with peaks 1, 3, 5, 9, 10, 12, 14, 16, 17, 18, and 19; PC2 shows a positive correlation with peaks 22, 24, and 25; PC3 shows a positive correlation with peaks 7 and 26 (Figure 5).

The factor scores obtained directly by SPSS are output as new variables. They were multiplied by the arithmetic square roots of the eigenvalues of the three principal component factors (Table 7). By this method, the scores of the three principal component factors were obtained and are listed in Table 8. The scores plot of PCA was shown in Figure 6. The C1 was significantly far away from the others, which could be confirmed by the lower similarity values mentioned in “2.6. HPLC Fingerprint Analysis”. Thus, it was notable that all samples were classified into three groups, including group I (C1), group II (C2, C3, and C4), and group III (C5, C6, C7, C8, C9, and C10). Although 10 batches of Morindae officinalis radix were collected from Zhaoqing City, Guangdong Province, they were clustered in different groups, which implied that PCA enabled quality control of Morindae officinalis radix samples, even if they were growing under the same geography condition. However, it still needed to be further investigated with more samples from different localities to reveal the relationship of growing environment with the corresponding quality of Morindae officinalis radix samples.

## 3. Materials and Methods

### 3.1. Materials

Ten batches of Morindae officinalis radix were collected from Zhaoqing City, Guangdong Province of China and authenticated as the roots of *Morinda officinalis* How by Associate Professor Xinjun Xu (School of Pharmaceutical Sciences, Sun Yat-sen University) (Table 9).

### 3.2. Reagents

HPLC-grade acetonitrile was purchased from Burdick & Jackson Labs (Honeywell B&J, WHNA, Muskegon, MI, USA). Absolute alcohol was of analytical grade and was obtained from Guanghua Technology Co., Ltd. (Guangzhou, China). Reference substances of sucrose, kestose, and nystose were provided by Anpel Laboratory Technologies Co., Ltd. (Shanghai, China).

### 3.3. Preparation of Standard Solutions

The mixed standard solution (10.70 mg/mL of sucrose, 10.30 mg/mL of kestose, and 10.2 mg/mL of nystose) was prepared by dissolution in water. A series of working solutions were prepared by diluting the standard solution with water to yield another five concentrations for the establishment of the calibration curve. All stock and working standard solutions were stored at 4 °C until they were used for analysis.

### 3.4. Preparation of Sample Solutions

Each of the crude Morindae officinalis radix powdered samples (1 g) were accurately weighed and soaked in 30 mL of 60% ethanol for 12 h, extracted by refluxing for 3 h at 90 °C, filtered, concentrated and transferred to 10 mL volumetric flasks. After dissolution with water, the extract was filtered through a 0.22 μm membrane filter and injected into the HPLC system for analysis.

### 3.5. HPLC Condition

HPLC analysis was performed on a LC-20AT HPLC system (Shimadzu, Japan) equipped with a LT II evaporative light scattering detector (ELSD). The chromatographic separation was performed using a COSMOIL Sugar-D column (4.6 mm × 250 mm, 5 μm, COSMOIL, Japan), operated at 35 °C. The mobile phase consisted of water (A) and acetonitrile (B) using a gradient program of 88–50% B (0–76 min) and 88% B (76–90 min). The flow rate and the injection volume were set at 1.0 mL/min and 20 μL. The drift tube temperature was 35 °C, and the gain was 4.

According to the guideline of International Conference on Harmonization (ICH), the developed method was validated for its system adaptability, linearity, repeatability, stability, and accuracy.

### 3.6. Tims TOF Conditions

Analysis was carried out on a UltiMate3000 HPLC system with Trapped-Ion-Mobility (tims) Time-Of-Flight (Bruker, Karlsruhe, Germany). Samples were operated on a Hypersil GOLD HILIC column (2.1 mm × 100 mm, 1.9 μm, Thermofisher, Waltham, MA, USA) with a column temperature of 40 °C. The mobile phase consisted of 0.1% formic acid aqueous solution (A) and acetonitrile (B) using a gradient program of 70% B (0–1 min), 70–45% B (1–15 min), 45–10% B (15–16 min), and 10% B (16–18 min). The flow rate was set at 0.3 mL/min.

### 3.7. Statistical Analysis

Statistical analysis was applied to demonstrate the variability of the 10 batches of Morindae officinalis radix samples. Software named Similarity Evaluation System for Chromatographic Fingerprint of Traditional Chinese Medicine (Version 2012.130723, Chinese Pharmacopoeia Commission, Beijing, China) was used for the similarity analysis according to the HPLC-ELSD data. Principal component analysis (PCA) was performed on the common chromatographic peaks in the HPLC fingerprints using the IBM SPSS Statistics software (IBM, Version 25, New York, NY, USA). 

## 4. Conclusions

In this study, the optimum extracting process of the saccharides in Morindae Officinalis Radix was determined by an orthogonal array design. The saccharides were extracted via 60% ethanol through heating reflux, avoiding the effect of low pH that led to the degradation of polysaccharides. The method showed good repeatability and accuracy, being the basis of the whole study. A powerful and reliable HPLC-ELSD method after validation using a new-type sugar-based column was developed for the quality assessment of the roots of *Morinda officinalis* How for the first time. The data of the method validation of HPLC fingerprint analysis showed that the method has good adaptability and stability in the study of chemical fingerprints of saccharides in the roots of *Morinda officinalis* How. The proposed method combined chemical fingerprint profiling of the herb and quantitative analysis of three major oligosaccharide components in the herb, i.e., sucrose, kestose, and nystose; HPLC-TIMS-TOF MS was further used to rapidly identify 17 saccharides (DP2-DP18) in the herb. The similarities of the roots were above 0.969. The results indicate that Morindae officinalis radix from different regions shared a similar HPLC pattern; all of them contained 26 characteristic peaks and possessed high concentrations of three oligosaccharide components but differed in content. Based on the fingerprints, 10 batches of Morindae officinalis radix were objectively classified by the multivariate statistical method (PCA). The study demonstrated that the developed method was efficient and reliable, which could be readily utilized as a more significant tool than the current one for the comprehensive quality control of Morindae officinalis radix. The operability and the analytical capacity offered by the developed HPLC-ELSD and HPLC-TIMS-TOF MS methods enabled their adoption as powerful analytical fingerprint techniques. Therefore, this HPLC fingerprint analysis, with better application prospects and broader application space, was very reliable in the control of the quality of Morindae officinalis radix.

## Figures and Tables

**Figure 1 molecules-26-07242-f001:**
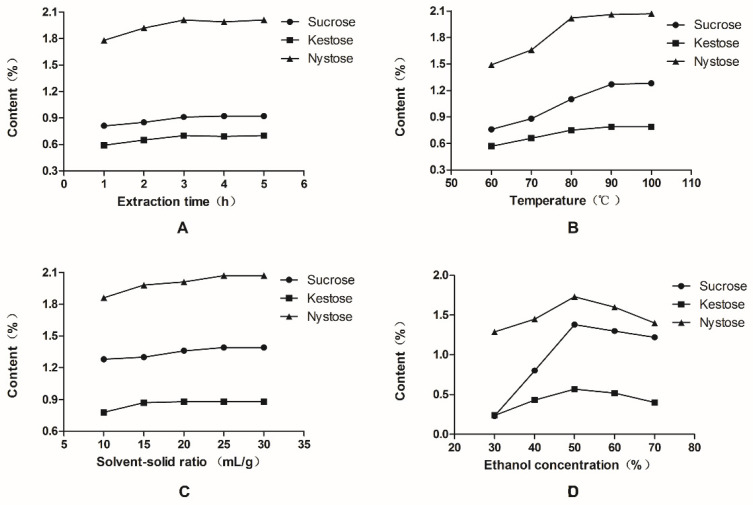
Effects of extraction conditions on the contents of sucrose, kestose, and nystose: (**A**) extraction time; (**B**) temperature; (**C**) solvent–solid ratio; (**D**) ethanol concentration.

**Figure 2 molecules-26-07242-f002:**
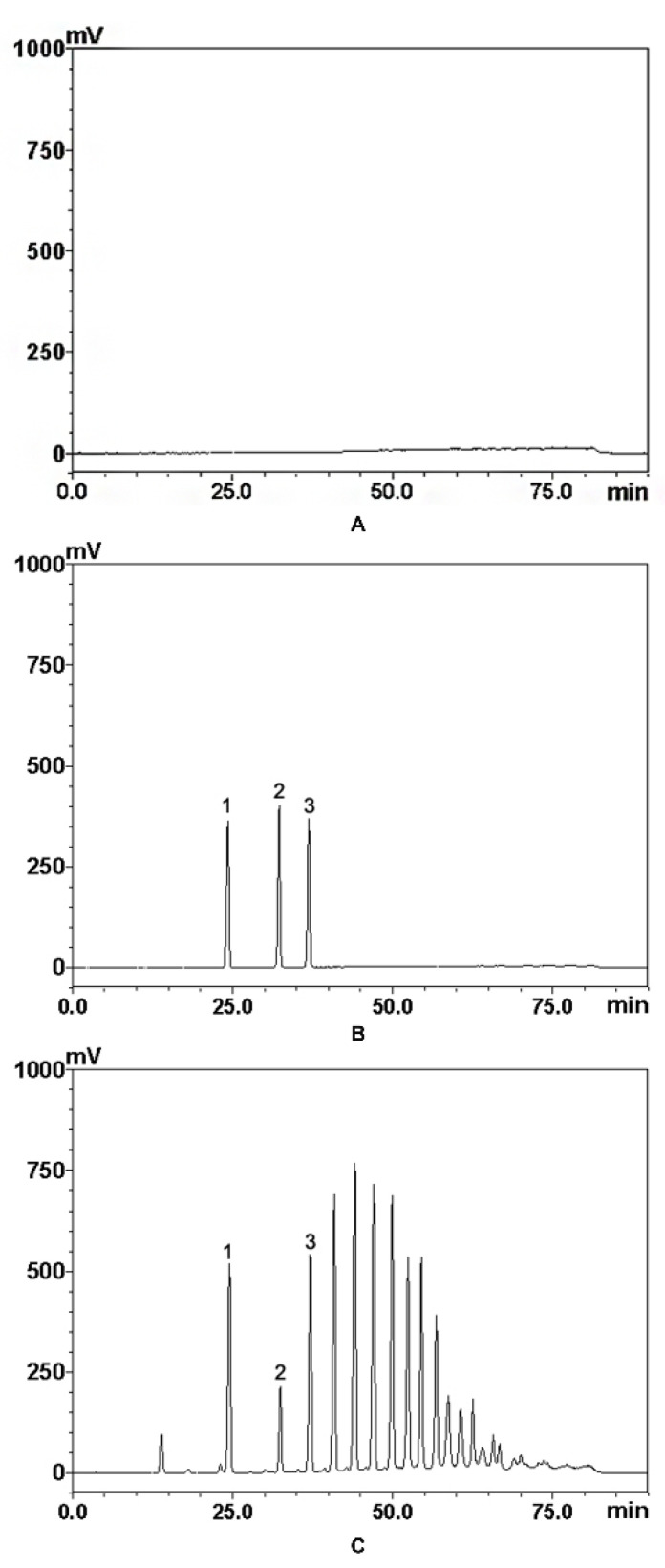
The chromatogram of blank (**A**), standards (**B**) and sample (**C**): 1 sucrose; 2 kestose; 3 nystose.

**Figure 3 molecules-26-07242-f003:**
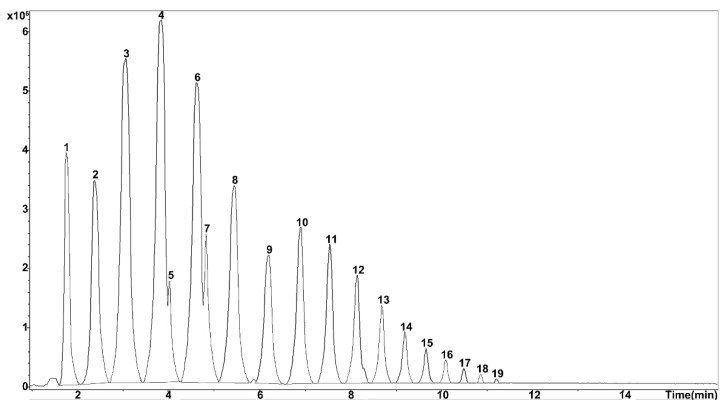
Total ion chromatogram of the root of *Morinda officinalis* How.

**Figure 4 molecules-26-07242-f004:**
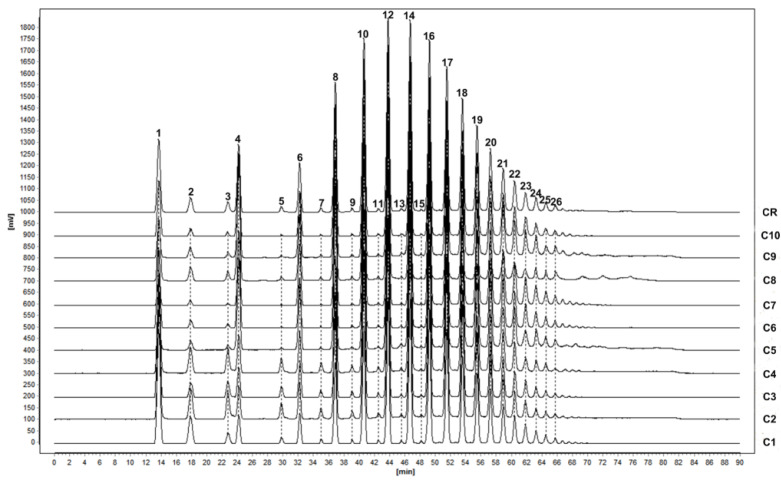
HPLC chromatographic fingerprints of 10 Morindae officinalis radix samples.

**Figure 5 molecules-26-07242-f005:**
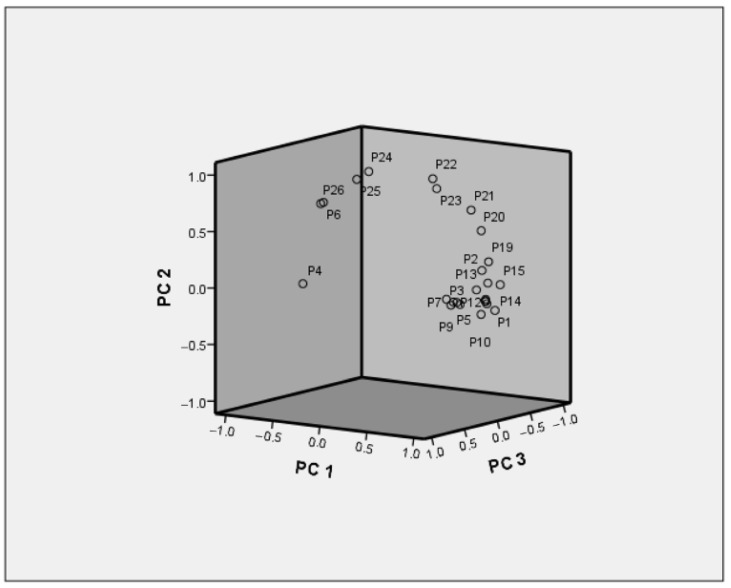
Loading plot of PCA.

**Figure 6 molecules-26-07242-f006:**
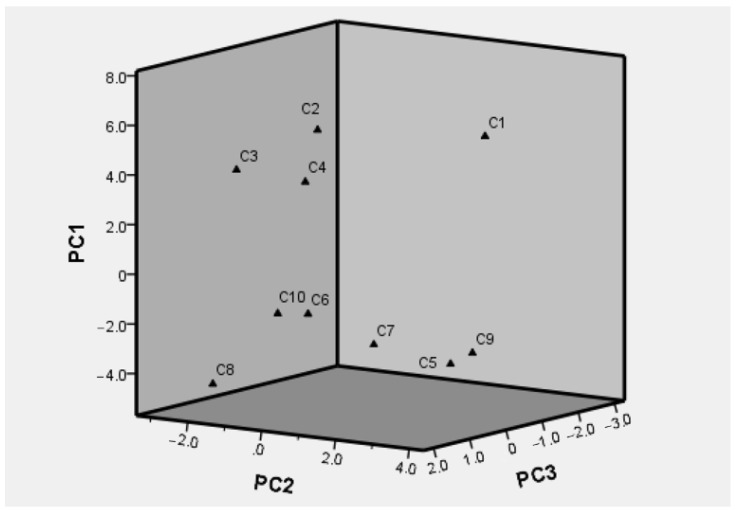
Scores plot of PCA.

**Table 1 molecules-26-07242-t001:** Orthogonal array design.

Level	Factor		
	A Temperature (°C)	B Solvent–Solid Ratio (mL/g)	C Ethanol Concentration (%)
1	80	20	40
2	90	25	50
3	100	30	60

**Table 2 molecules-26-07242-t002:** Orthogonal test results.

No.	A Temperature	B Solvent–Solid Ratio	C Ethanol Concentration	Sucrose (%)	Kestose (%)	Nystose (%)	Overall Desirability
1	2	1	3	1.57	0.76	2.11	0.91
2	2	3	2	1.17	0.64	1.91	0.51
3	1	3	3	1.68	0.71	2.07	0.85
4	3	3	1	0.84	0.62	1.85	0.30
5	1	1	1	0.73	0.58	1.71	0.08
6	2	2	1	1.08	0.77	2.06	0.66
7	3	1	2	1.09	0.61	1.61	0.13
8	3	2	3	1.09	0.52	1.72	0.13
9	1	2	2	1.11	0.65	1.78	0.42
k1	0.45	0.37	0.35				
k2	0.69	0.40	0.36				
k3	0.19	0.55	0.63				
R	0.51	0.18	0.28				

**Table 3 molecules-26-07242-t003:** Linear regression data, LODs and LOQs of sucrose, kestose, and nystose.

Compo-nent	Regression Equation	r	Linear Range (mg/mL)	LOQ (μg/mL)	LOD (μg/mL)
Sucrose	LgY = 1.127 lgX + 3.114	0.9995	0.67–9.36	2.5	1.1
Kestose	LgY = 1.102 lgX + 3.2269	0.9998	0.64–9.01	3.1	1.4
Nystose	LgY = 0.962 lgX + 3.7433	0.9995	0.64–8.93	4.8	2.1

**Table 4 molecules-26-07242-t004:** System adaptability, repeatability, stability, and recovery of sucrose, kestose, and nystose.

Component	System Adaptability (*n* = 6)	Repeatability (*n* = 6)	Stability (*n* = 7)	Recovery (*n* = 9)	
	RSD (%)	RSD (%)	RSD (%)	Mean (%)	RSD (%)
Sucrose	2.53	1.09	3.06	99.12	3.59
Kestose	1.57	2.94	2.75	101.72	1.26
Nystose	1.21	2.75	2.25	96.70	2.24

**Table 5 molecules-26-07242-t005:** Identification of oligosaccharides from the root of *Morinda officinalis* How.

No.	t_R_ (min)	[M-H]^−^/[M-2H]^2−^	Error (ppm)	Molecular Weight	Molecular Formula	Identification
1	1.8	341.1090	0.29	342.1162	C_12_H_22_O_11_	Sucrose
2	2.4	503.1618	0.00	504.1690	C_18_H_32_O_16_	Kestose
3	3.1	665.2155	1.35	666.2218	C_24_H_42_O_21_	Nystose
4	3.9	827.2675	0.12	828.2747	C_30_H_52_O_26_	Fructofuranosylnystose
6	4.6	989.3186	−1.62	990.3275	C_36_H_62_O_31_	Fructooligosaccharide (DP6)
8	5.5	1151.3694	−3.21	1152.3803	C_42_H_72_O_36_	Fructooligosaccharide (DP7)
9	6.2	1313.4215	−3.35	1314.4332	C_48_H_82_O_41_	Fructooligosaccharide (DP8)
10	6.9	1475.4748	−2.64	1476.4860	C_54_H_92_O_46_	Fructooligosaccharide (DP9)
11	7.6	1637.5290	−1.53	1638.5388	C_60_H_102_O_51_	Fructooligosaccharide (DP10)
12	8.2	1799.5829	−0.78	1800.5916	C_66_H_112_O_56_	Fructooligosaccharide (DP11)
13	8.7	980.3144	−0.51	1962.6444	C_72_H_122_O_61_	Fructooligosaccharide (DP12)
14	9.2	1061.3396	−1.70	2124.6973	C_78_H_132_O_66_	Fructooligosaccharide (DP13)
15	9.7	1142.3656	−1.93	2286.7501	C_84_H_142_O_71_	Fructooligosaccharide (DP14)
16	10.1	1223.3907	−2.86	2448.8029	C_90_H_152_O_76_	Fructooligosaccharide (DP15)
17	10.5	1304.4175	−2.38	2610.8557	C_96_H_162_O_81_	Fructooligosaccharide (DP16)
18	10.9	1385.4434	−2.60	2772.9085	C_102_H_172_O_86_	Fructooligosaccharide (DP17)
19	11.2	1466.4713	−1.43	2934.9613	C_108_H_182_O_91_	Fructooligosaccharide (DP18)

**Table 6 molecules-26-07242-t006:** The contents of sucrose, kestose, and nystose in 10 samples of Morindae officinalis radix (*n* = 3).

Sample	Sucrose (%)	Kestose (%)	Nystose (%)
C1	1.21 ± 0.03	1.02 ± 0.02	2.38 ± 0.02
C2	1.50 ± 0.03	1.37 ± 0.02	3.45 ± 0.05
C3	1.21 ± 0.03	1.24 ± 0.03	3.10 ± 0.04
C4	1.71 ± 0.04	1.50 ± 0.03	3.72 ± 0.10
C5	3.44 ± 0.06	2.44 ± 0.05	4.99 ± 0.14
C6	3.84 ± 0.10	1.74 ± 0.05	4.51 ± 0.05
C7	2.86 ± 0.05	1.97 ± 0.04	4.45 ± 0.11
C8	4.56 ± 0.02	1.83 ± 0.00	3.98 ± 0.04
C9	3.32 ± 0.06	2.22 ± 0.01	4.89 ± 0.04
C10	2.91 ± 0.06	1.55 ± 0.03	3.93 ± 0.03

**Table 7 molecules-26-07242-t007:** Eigenvalue and variance contribution rate of three main component factors.

	Eigenvalue	Variance Contribution Rate, %	Cumulative Variance Contribution Rate, %
PC1	16.429	65.715	65.715
PC2	5.215	20.859	86.574
PC3	1.813	7.250	93.824

**Table 8 molecules-26-07242-t008:** Scores of three principal component factors.

Sample	Score of PC1	Score of PC2	Score of PC3
C1	4.658	1.298	−2.573
C2	6.104	0.426	1.164
C3	3.438	−2.970	−0.059
C4	4.254	0.648	1.732
C5	−3.067	3.312	0.432
C6	−2.442	−1.801	−0.844
C7	−3.232	0.183	−0.638
C8	−4.302	−1.791	1.798
C9	−2.816	3.358	−0.127
C10	−2.595	−2.664	−0.886

**Table 9 molecules-26-07242-t009:** Sources of samples.

Source	Collected Date	The Roots
Gaoliang village	2018.10	R1
Dazhai village	2018.10	R2
Zhongxiong village	2018.10	R3
Luoyang village	2018.10	R4
Guancun village	2018.10	R5
Shashui village	2018.10	R6
Dajiang village	2018.10	R7
Jinshan village	2018.10	R8
Wanxing village	2018.10	R9
Yunli village	2018.10	R10

## Data Availability

Data sharing not applicable.

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
