# Peer review of "Chemical Fingerprint Analysis and Quantitative Analysis of Saccharides in Morindae Officinalis Radix by HPLC-ELSD"

_molecules, 2021, doi:10.3390/molecules26237242_

Round 1

Reviewer 1 Report

In my opinion, the manuscript titled ,,Chemical Fingerprint Analysis and Quantitative Analysis of Saccharides in Morindae Officinalis Radix by HPLC-ELSD’’ can be recommended for publication in Molecules, however after minor revision.

In this manuscript, the authors developed a high performance liquid chromatography with an evaporative light scattering detection method (HPLC-ELSD) for the determination of three active compounds and chemical fingerprint analysis of saccharides in Morindae officinalis radix. Trapped-Ion-Mobility tims Time-Of-Flight (tims TOF) was also used to identify saccharides of Morindae officinalis radix. The structure of the article is rather well organized and well balanced. There is a clear indication of the objective, scope, and results of the paper.

My remarks and recommendations are as follows:

  1. ,,Method Validation of Quantitative Analysis’’ - please add information: the method calculation of recovery,  the range of the standard curves, the limit of detection and limit of quantitation (LOD and LOQ,  with information about the calculation method).
  2. Additionally, the chemical formulas of the determined compounds (sucrose, kestose and nystose) can be presented.
  3. There are some typos in the text. The text of manuscript should be carefully checked.

Reviewer 2 Report

This manuscript presents a study about a novel method based on high performance liquid chromatography with an evaporative light scattering detection (HPLC-ELSD) for quantitative analysis of three active compounds and chemical fingerprint analysis of saccharides in Morindae officinalis radix.

To this end, the samples were separated with a COSMOIL Sugar-D column (4.6 mm×250 mm, 5 μm) by using gradient elution with water (A) and acetonitrile (B). In addition, Trapped-Ion-Mobility tims Time-Of-Flight (tims TOF) was used to identify saccharides of Morindae officinalis radix.

From my inspection of the manuscript, I find the manuscript suitable to be published in Molecules after a major and very deep revision. My main comments are as follows:

(1) Introduction: the introduction gives light in the two main aspects that should be considered in this part of the paper: the importance of the analyte and a mini "review" of the analytical tehcnicals previosly used for similar studies. This last information reinforces the convenience of using the method proposed by the authors.

(2) Materials and methods: this section is, in general, poor. A more descriptive section is needed to understand what are the authors doing in their experiments and, more interestingly, make their experiments to be reproducible. Actually, it is not possible to do it.

-Why the pH was not considered in the optimization of the extraction conditions? Surely the pH could play an important role when changing from acidic to basic conditions. In addition, no information about the pH of all the solutions in the manuscript can be found. 

- No information about the factorial design was provided. Why did you use an orthogonal array design? Did you consider the information lost with your choice?

(3) Results: The results are presented, there are no discussion about it. No references to other published works are included. Therefore, it is difficult to assess how this manuscript presents good or not interesting results.

(4) Conclusions: based on my previous comments, the conclusions, actually, are not supported by the results.

Round 2

Reviewer 2 Report

I suggested the authors to make a deep revision of their manuscript. However, no responses to my comments have been considered to improve the manuscript.

My main concerns against the acceptance of this manuscript in its actual form actually maintains in the same terms as they were described in my first revision.

The authors have commented and improved their manuscript based on the suggestions of reviewer 1, but they have not considered those mine. This journal maintains a high quality based on the manuscript revisions by pairs. The authors have not considered important this critical observation. Therefore I can only recommend to reject the article in its actual form.
